# Using targeted sequencing and TaqMan approaches to detect acaricide (bifenthrin, bifenazate, and etoxazole) resistance associated SNPs in *Tetranychus urticae* collected from peppermint fields and hop yards

Silas Shumate[1], Maggie Haylett[1], Brenda Nelson[1], Nicole Young[1], Kurt Lamour[2], Doug Walsh[3], Benjamin Bradford[4], Justin Clements[1]*

1 Department of Entomology, Plant Pathology, and Nematology, University of Idaho, Parma, Idaho, United States of America, 2 Department of Genome Science and Technology, University of Tennessee, Knoxville, Tennessee, United States of America, 3 Department of Entomology, Washington State University, Prosser, Washington, United States of America, 4 Department of Entomology, University of Wisconsin-Madison, Madison, Wisconsin, United States of America

* justinclements@uidaho.edu

## Abstract

*Tetranychus urticae* (Koch) is an economically important pest of many agricultural commodities world-wide. Multiple acaricides, including bifenazate, bifenthrin, and extoxazole, are currently registered to control *T. urticae*. However, populations of *T. urticae* in many different growing regions have developed acaricide resistance through multiple mechanisms. Within *T. urticae*, single nucleotide polymorphisms (SNPs) have been documented in different genes which are associated with acaricide resistance phenotypes. The detection of these mutations through TaqMan qPCR has been suggested as a practical, quick, and reliable tool to inform agricultural producers of acaricide resistance phenotypes present within their fields and have potential utility for making appropriate acaricide application and integrated pest management decisions. Within this investigation we examined the use of a TaqMan qPCR-based approach to determine genotypes which have been previously associated with acaricide resistance in field-collected populations of *T. urticae* from peppermint fields and hop yards in the Pacific Northwest of the United States and confirmed the results with a multiplex targeted sequencing. The results suggest that a TaqMan qPCR approach accurately genotypes *T. urticae* populations for SNPs that have been linked to Bifenazate, Bifenthrin, and Etoxazole resistance. The results also demonstrated that different populations of mites in Washington and Idaho displayed varying frequencies of the examined SNPs. While we were able to detect the SNPs associated with the examined acaricides, the mutation G126S was not an appropriate or accurate indicator for bifenazate resistance.

**Data Availability Statement:** All relevant data are within the paper and its Supporting information files.

**Funding:** This research was supported by funding from United States Department of Agriculture Specialty Crop Initiative 2021-51181-35901, awarded to DW and JC, a Mint Industry Research Council grant awarded to DW and JC, a Washington Mint Commission grant awarded to DW, and startup funds awards from the University of Idaho to JC. There was no additional external funding received for this study. The funders had no role in study design, data collection and analysis, decision to publish, or preparation of the manuscript.

**Competing interests:** The authors have declared that no competing interests exist.

## Introduction

*Tetranychus urticae* (Koch) is an economically relevant pest of multiple agricultural commodities in the Pacific Northwest, including hops and mint [1, 2]. *Tetranychus urticae* are small (0.5 mm) mites that predominantly live and feed on the underside of plant leaves [3]. This species is incredibly adaptable and can feed on more than 1,000 different plant species, including multiple cropping systems found in Idaho and Washington [2, 4]. The gnathosoma of *T. urticae* is complex, with chelicera that have been adapted to needle-like mouthparts for piercing and sucking of plant material, removing chlorophyll from plant cells, and causing extensive damage [5–7]. The damage caused by *T. urticae* can result in necrotic spots, yellowing, loss of leaves that are fed upon, and consequently reduce photosynthesis [6, 8]. In both hops and mint, feeding injury resembles water stress. High infestations late in the growing season in the Pacific Northwest can result in economic loss of crops, including mint stands and hop yards. In hops *T. urticae* can be a direct pest to hop cones, which can cause off colors (less green) or, in the case of severe infestation, can cause brittleness and shattering of the cone during mechanical harvest. These defects can result in growers having their crop rejected by hop merchants or breweries. As such, growers rely on both cultural integrated pest management practices and acaricide compounds to control populations of *T. urticae* [9].

To date numerous populations of *T. urticae* have developed resistance to over 96 different acaricide chemistries, with over 140 publications documenting the development of resistance [10]. Acaricide resistance has been documented world-wide [10]. Currently, agricultural producers proactively monitor pest populations through arthropod scouting for pest population thresholds and apply foliar acaricides when established thresholds are reached [11]. If the acaricide application is ineffective, researchers and crop consultants can screen pest populations for acaricide resistance using median lethal dose assays ($LD_{50}$) or by sending pest populations to extension agencies or pest scouting agents to be screened. However, these approaches can take weeks to determine resistant phenotypes. Instead, the grower will usually make an informed decision and apply another pesticide treatment for control. Unfortunately, using the incorrect chemical or wrong concentration of the correct chemical may not provide adequate control and/or can further drive the development of acaricide resistance [11].

Acaricide resistance can develop through multiple mechanisms, including enhanced metabolic breakdown of acaricides, target site insensitivity, and behavioral resistance [12, 13]. Acaricide resistance is not unique to *T. urticae*, and other mite species, including *Dermanyssus gallinae*, *Panonychus citri*, *Panonychus ulmi*, have demonstrated resistance to acaricides through biochemical and molecular mechanisms of acaricide detoxification [14, 15]. One of the most important mechanisms for acaricide resistance development in *T. urticae* is target site insensitivity, where single nucleotide polymorphisms (SNPs) can result in an alteration of an amino acid sequence and the translated protein which, in-turn, binds the insecticide more weakly or not at all [13]. These polymorphisms are non-synonymous and may be the result of natural selection when they result in beneficial traits [16]. SNPs in multiple genes have become associated with the development of resistance to different acaricide chemistries, including bifenthrin, bifenazate, and etoxazole [17–20]. Historically, these mutations have been detected through DNA sequencing that can be analyzed for the presence of the polymorphism. Multiple investigations have recently suggested using these polymorphisms as a diagnostic tool to inform agricultural growers of the acaricide resistance status of individual field populations of *T. urticae*. This would allow the grower to make an informed decision into which acaricide would adequately control their local *T. urticae* populations and may prevent further resistance development [21, 22]. Three of the most common polymorphisms studied are a mutation that results in an amino acid change. The first results in a change of a glycine to a serine in the

cytochrome b gene (G126S) in the mitochondrial respiratory chain. This mutation has been shown to confer resistance to bifenazate, a carboxylic ester, in *T. urticae* [23]. However, the effects of the G126S mutation have recently been called into question and new research suggests that this mutation cannot be used to predict acaricide resistance [24]. Another mutation at amino acid 1538 in the voltage gate channel gene results in a phenylalanine changing to an isoleucine. This mutation has been suggested to confer resistance to bifenthrin [19], a pyrethroid, and has also been shown to confer resistance to the growth inhibitors, hexythiazox and clofentezine [17]. Some studies bring into question the accuracy of F1538I to predict bifenthrin resistance [25]. Finally, a mutation in the chitin synthase 1 gene that changes an isoleucine to a phenylamine has been shown to result in resistance to etoxazole, a narrow spectrum systemic acaricide [20]. These mutations have been characterized by sequencing both susceptible and resistant populations of *T. urticae* and have provided in-depth information on the development of resistance to acaricides in *T. urticae* [16–20, 23]. A restriction digest has also been used to screen mite populations for the I1017F mutation [26].

One potential approach for detecting resistance associated SNPs is to use a TaqMan quantitative PCR genotyping assay [21, 22]. This approach allows for the detection of SNPs using two fluorescent probes; one labeled to detect the wildtype allele and the other to detect the allele with the mutation. Historically, TaqMan based approaches have been used to examine allelic frequency within a single individual [21, 22]. The presence of the mutation can be quantified within a DNA sample and can be used to classify an individual within a population as wildtype (susceptible), heterozygous (susceptible/resistant), or resistant phenotype. This information can be used to predict the efficacy of an acaricide prior to an application. However, screening hundreds of individuals mites within a population is not a cost-effective or practical monitoring solution for a grower. Additionally, recent findings indicate that some of the polymorphisms associated with resistance may not be useful as a diagnostic tool, including G126S in the cytochrome b gene and F1538I in the voltage gate channel [24, 25]. In the present study, we set out to investigate whether we could use a pooled TaqMan qPCR approach to detect resistance associated SNPs in field populations of *T. urticae* from mint fields and hop yards in the Pacific Northwest of the United States. We chose to examine a limited set of three well defined markers G126S (Cytochrome b), F1538I (Voltage gated sodium channel), and I1017F (Chitin synthase 1) using a pooled TaqMan approach. We examined allelic frequency and confirmed the findings of the TaqMan assay with a multiplex PCR. Our results indicate that a TaqMan qPCR-based approach has the potential to quickly identify populations of *T. urticae* that possess resistance associated SNPs.

## Materials and methods

### Ethics statement

This article does not contain studies with any human participants and no specific permits were required for collection or experimental treatment of *Tetranychus urticae* for the study described.

### *Tetranychus urticae* collections

*Tetranychus urticae* were collected from mint fields and hop yards in the Pacific Northwest during the summer of 2021 (Fig 1). Briefly, *T. urticae* infested leaves were hand collected, placed in a sealed plastic bag, and transported in a cooler back to the Parma Research and Extension Center, Parma, Idaho or the Irrigated Agriculture Research and Extension Center (IAREC), Prosser, Washington. Samples were shared between research stations. A total of thirteen populations of *T. urticae* were collected from Washington and Idaho (9 populations from

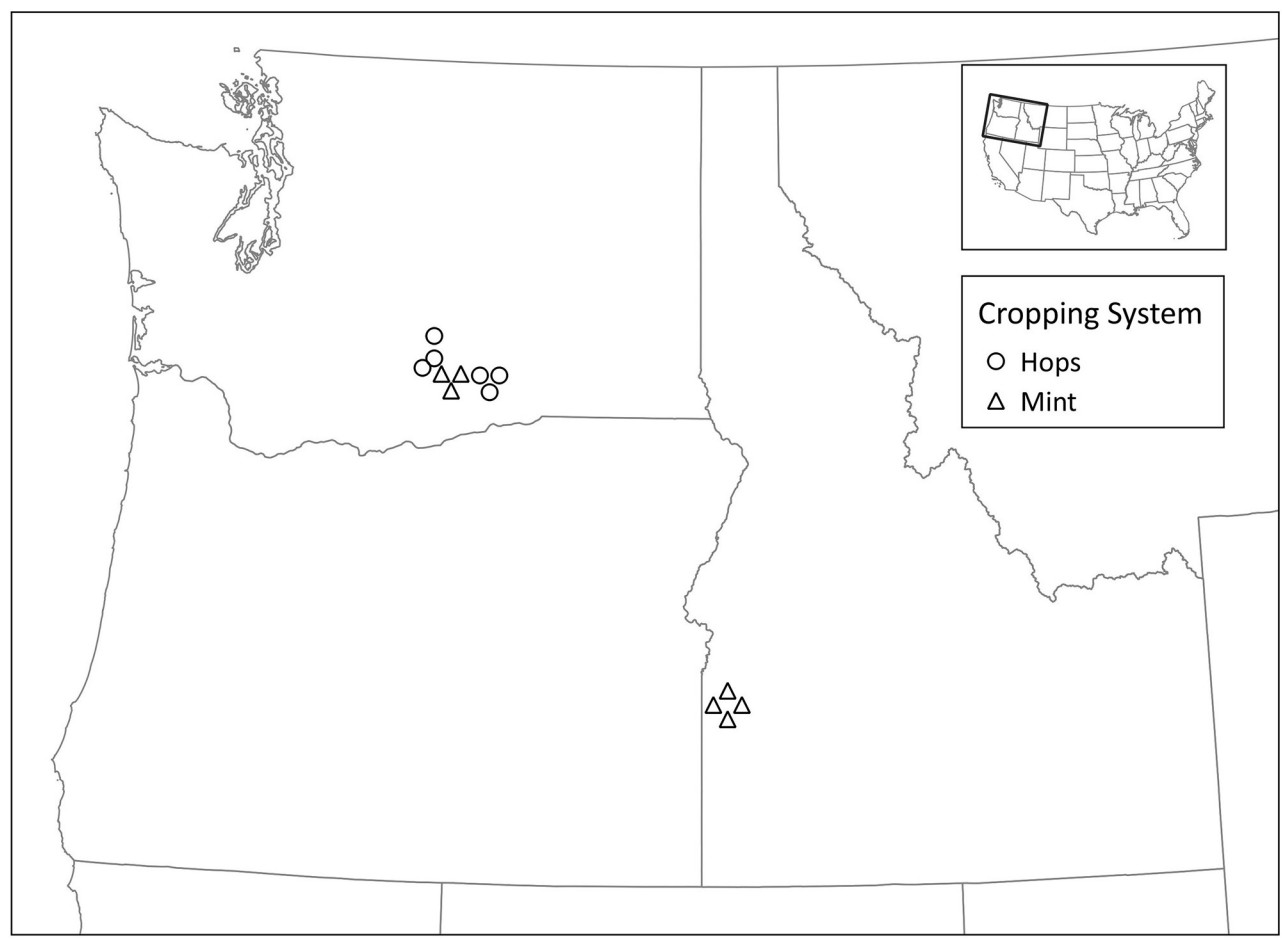

**Fig 1. Field collection sites for *T. urticae* from mint fields and hop yards in the Pacific Northwest of the United States.**

Washington and 4 populations from Idaho). Samples were placed in insect/mite proof mesh cages and housed on beans while DNA extraction and median lethal dose assays were conducted.

## DNA isolation from *Tetranychus urticae*

*Tetranychus urticae* DNA was isolated using two different methods; the first was a modified Cethyl Trimethyl Ammonium Bromide (CTAB) method (OPS Diagnostics, NJ, USA) and the second was a Zymo Research Quick-DNA Tissue/ Insect Microprep kit (Zymo Research, CA, USA). Heavily infested *T. urticae* leaves were removed from sealed cages and placed in a sealed petri dish to eliminate movement of mites. Mites were brushed from leaves with a mite brushing machine (Leedom Engineering, CA, USA) onto glasses plates. The mite brushing machine was cleaned before and after use and sterilized with 254 nm UV light for 15 minutes. After samples of mites were collected, the glass plates were chilled at 4˚C for 20 mins. The plate was observed under a dissecting microscope to confirm mite species and remove any possible contaminants. Mite samples were then scraped into 2 ml polypropylene tubes using two sterile razor blades. This method of removing the mites from the leaves was done for both extraction methods.

DNA extractions were conducted on pooled motile mites encompassing multiple life stages. We collected as many mites as possible from each population and the average extraction size was approximately 275 mites per population. For the modified CTAB extraction protocol, aggregate samples of mites were placed in 2 ml DNAse/RNAse-free homogenate tubes (Biospec, OK, USA) with a single sterilized 6.4 mm and four 2.7 mm diameter glass bead (Biospec, OK, USA). Seven hundred and fifty µl of CTAB Extraction Buffer was added to each tube (OPS diagnostics, NJ, USA). Samples were homogenized for 2 minutes in a Mini-beadbetter-16 (Biospec, OK, USA). Tubes with homogenate were incubated at 60 ˚C in a water bath for 30 minutes. Following the incubation period, samples were centrifuged for 10 minutes at 14,000 x g and the supernatant was transferred to a new 1.5 ml tube. Five µl of RNase solution A (20 mg/ml, Fisher Scientific, MA, USA) was added and incubate at 37 ˚C for 20 minutes. Three hundred µl of chloroform/isoamyl alcohol (24:1) was added to each sample and vortexed for 5 seconds and samples were then centrifuged for 1 minute at 14,000 x g to separate the phases. The chloroform/isoamyl alcohol step was conducted twice. The upper aqueous phase was transferred to a new 1.5 ml microcentrifuge tube. DNA was precipitated by adding 500 µl cold isopropanol. Samples were left for 12 hours at -20 ˚C. Samples were then centrifuged at 14,000 x g for 10 minutes to pelletize DNA. Supernatant was decanted and discarded without disturbing the pellet, which was subsequently washed with 1 ml of ice cold 70% ethanol, and then vortexed and centrifuged at 14,000 x g for 10 minutes. Ethanol was decanted and excess ethanol was removed with a pipettor from the pellet. Samples were air dried in a sterile PCR cabinet for 15 min. DNA was dissolved in 100 µl RNase/DNase free $H_2O$. DNA concentration was determined using a Nanodrop 2000. Samples were stored at -20 ˚C until multiplex and TaqMan processing. For the Zymo Research Quick-DNA Tissue/ Insect Microprep kit extraction method, aggregate samples of mites were added to ZR BashingBead Lysis Tube (2.0 mm) and 750 µl BashingBead buffer. The mites within the tubes were then homogenized for 6 minutes, after which Zymo Research Quick-DNA Tissue/ Insect Microprep kit methodology was followed, and DNA was eluted from the spin column with 20 µl of DNA Elution Buffer. *Tetranychus urticae* DNA concentration was determined using a Nanodrop 2000. Samples were stored at -20 ˚C until multiplex PCR and TaqMan qPCR processing.

## Genotyping quantitative PCR

TaqMan quantitative PCR was used to examine the presence of different alleles within populations of *T. urticae* for mutations in Cytochrome b (G126S) [23], Voltage gated sodium channel (F1538I) [19], and Chitin synthase 1 (I1017F) [20]. Accession numbers for each reference gene can be found in S1 Table. TaqMan based primers and probes were designed and purchased through Custom TaqMan SNP Genotyping Assays Designs Tool (Thermo Fisher, MA, USA). Primer and probes can be found in S1 Table. The TaqMan-based assay uses both FAM and VIC fluorescent probes, with the wildtype allele labeled with the VIC probe and the FAM probe for the mutation that is associated with the resistant allele. The TaqMan qPCR reaction was conducted in 25 µl reactions using the Applied Biosystems TaqMan Genotyping Master Mix (Applied Biosystems, MA, USA) and followed recommended procedures. Duplicate reactions were run at 95 ˚C for 10 min, followed by 95 ˚C for 15 s, and 60 ˚C for 60 s for a total of 40 cycles on a CFX 96 Bio-Rad machine (Applied Biosystems, MA, USA). Data was analyzed using Bio-Rad CFX Maestro Software (Applied Biosystems, MA, USA) to group populations into wildtype, heterozygous, and resistance associated genotypes. Unknown calls were mapped to the closest genotype. Negative template controls were run along with all samples as outline in Bio-Rad CFX software manual (Applied Biosystems, MA, USA).

## Multiplex targeted-sequencing

Total DNA from each sample was sent to Floodlight Genomics, LLC. Floodlight Genomics used an optimized Hi-Plex approach to amplify and barcode targets in a single multiplex PCR reaction followed by sequencing on an Illumina HiSeqX device running a 2x150bp paired-end configuration, as previously described [27]. Primers were designed to amplify three 150 bp regions to examine known SNPs previously correlated with acaricide resistance in *T. urticae* (S1 Table), G126S (Cytochrome b) [23], F1538I (Voltage gated sodium channel) [19], and I1017F (Chitin synthase 1) [20]. The sample-specific barcoded amplicons were sequenced on the Illumina HiSeq X platform according to the manufacturer's directions. Floodlight Genomics delivered sample-specific raw DNA sequence reads as FASTQ files. Annotation of the raw reads was performed with Geneious Bioinformatics Software (Auckland, New Zealand). The reads were then mapped to reference sequences at 100% stringency to classify genotype. The genotype ratio was calculated for each population. Each population was designated with a wildtype or F15381/I1017Fgenotype if ≥90% of the reads corresponded to that associated genotype.

## Medium lethal dose assay

Median lethal dose bioassays were conducted on 1 cm bean leaf disks with 4 replicate leaf disks per rate tested. These mites were treated with acaricides at increasing rates. Bifenazate treatment rates were 0 (control), 3.12%, 6.25%, 12.5%, and 50% of the maximum label rates permitted of Vigilant 4SC (24 fluid oz per acre). Treatments were sprayed onto the leaf surface with a potter spray tower. Ten gravid adult female *T. urticae* were placed on each leaf disk. Each bioassay arena for each of the tested rates was treated with 2 ml of the dilute solution. After 24 hours mites, were evaluated for mortality. Mites were classified as dead when they failed to move more than their body length when prodded with a small paint brush. Moribund mites rarely recover and typically die within the next 24 hrs. The $LD_{50}$ (lethal dose that controls 50% of the population) the 95% CI (confidence interval), the slope ± SEM (standard error of the mean), $X^2$ and df were calculated with the software Polo Probit (LeOra Software LLC). A laboratory population of *T. urticae* was used as a baseline reference for the $LD_{50}$ assays. This population represents a historically susceptible population that has been housed at the IAREC center in Processer Washington for multiple generations, has not been subjected to any chemical exposure over those generations, and has been raised on beans.

## Results

### TaqMan quantitative PCR detection of bifenazate (G126S), bifenthrin (F1538I), and etoxazole (I1017F) resistance

The TaqMan PCR genotyping assay detected SNPs associated with bifenazate (G126S), bifenthrin (F1538I), and etoxazole (I1017F) resistance in samples of *T. urticae* collected in mint and hops in Washington and Idaho (Table 1). Within the populations that were examined, we observed that no population was complete fixed with the G126S mutation. Most populations collected from mint were identified as wildtype, while most populations from hops were heterozygous and composed of individuals with both wildtype and the G126S mutation. For bifenthrin (F1538I mutation), we observed most populations of mites collected were heterozygous. One population collected in mint from Idaho identified as wildtype and another population collected from mint in Idaho was fixed for the F1538I mutation. The mutation associated with etoxazole resistance (I1017F) was split regionally, with populations of mites collected in Idaho expressing wildtype genotype and populations of mites collected in Washington

**Table 1. TaqMan assay of *T. urticae* collected from hops and mint in the Pacific Northwest.**

| Pop. | Cropping System | Growing Region | G126S (Bifenazate) | F1538I (Bifenthrin) | I1017F (Etoxazole) |
|---|---|---|---|---|---|
| | | | TaqMan | TaqMan | TaqMan |
| 1 | Hops | Washington | Heterozygous | Heterozygous | I1017F |
| 2 | Hops | Washington | Heterozygous | Heterozygous | I1017F |
| 3 | Hops | Washington | Wildtype | Heterozygous | Heterozygous |
| 4 | Hops | Washington | Heterozygous | Heterozygous | Heterozygous |
| 5 | Hops | Washington | Heterozygous | Heterozygous | Heterozygous |
| 6 | Hops | Washington | Heterozygous | Heterozygous | Heterozygous |
| 7 | Mint | Idaho | Wildtype | Heterozygous | Wildtype |
| 8 | Mint | Idaho | Wildtype | Heterozygous | Wildtype |
| 9 | Mint | Idaho | Wildtype | F1538I | Wildtype |
| 10 | Mint | Idaho | Heterozygous | Wildtype | Wildtype |
| 11 | Mint | Washington | Wildtype | Heterozygous | Heterozygous |
| 12 | Mint | Washington | Wildtype | Heterozygous | Heterozygous |
| 13 | Mint | Washington | Wildtype | Heterozygous | Heterozygous |

*G126S/F1538I/I1017F and wildtype refer to a homozygous genotype in which the genotype is fixed with either the wildtype genotype or the associated SNP.

expressing heterozygous or fixed I1017F genotypes. However, the TaqMan assay could not determine the allelic frequency for the examined mutations, and instead only provided whether the population was wildtype, heterozygous, or a completely fixed mutant.

## Targeted-sequencing and detection of bifenthrin (F1538I,) and etoxazole (I1017F) resistance

A multiplex targeted-sequencing approach provided additional insight into the relative frequency of the SNPs within each population. The multiplex assay provides read counts that can be correlated to the mutation within a population (Table 2). The mutation associated with

**Table 2. Comparison between TaqMan qPCR and multiplex PCR results in *T. urticae* collected from hops and mint in the Pacific Northwest.**

| Pop. | Cropping System | Growing Region | F1538I (Bifenthrin) | | I1017F (Etoxazole) | |
|---|---|---|---|---|---|---|
| | | | TaqMan | Multiplex | TaqMan | Multiplex |
| 1 | Hops | Washington | Heterozygous | 1.49, 59.87% | I1017F | 56.46, 98.25% |
| 2 | Hops | Washington | Heterozygous | 2.36, 70.28% | I1017F | 4.73, 82.55% |
| 3 | Hops | Washington | Heterozygous | 0.87, 46.67% | Heterozygous | 2.63, 72.46% |
| 4 | Hops | Washington | Heterozygous | 2.07, 67.43% | Heterozygous | 1.21, 54.89% |
| 5 | Hops | Washington | Heterozygous | 2.33, 70.02% | Heterozygous | 1.33, 57.11% |
| 6 | Hops | Washington | Heterozygous | 0.57, 36.32% | Heterozygous | 0.45, 31.14% |
| 7 | Mint | Idaho | Heterozygous | 0.53, 34.72% | Wildtype | 0.093, 8.58% |
| 8 | Mint | Idaho | Heterozygous | 0.56, 35.90% | Wildtype | 0.034, 3.30% |
| 9 | Mint | Idaho | F1538I | 11.06, 91.70% | Wildtype | 0.0028, 0.28% |
| 10 | Mint | Idaho | Wildtype | 0.0034, 0.34% | Wildtype | 0.0049, 0.49% |
| 11 | Mint | Washington | Heterozygous | 1.13, 53.18% | Heterozygous | 0.44, 30.77% |
| 12 | Mint | Washington | Heterozygous | Did Not Amplify | Heterozygous | Did Not Amplify |
| 13 | Mint | Washington | Heterozygous | 1.47, 59.67% | Heterozygous | 0.94, 48.70% |

* Multiplex reads are presented as a ratio of resistance associated allele genotype over wildtype allele genotype and percent of resistance associated genotype reads.
*F15381/I1017F and wildtype refer to a homozygous genotype in which the genotype is fixed with either the wildtype genotype or the resistance associated SNP.

**Table 3. Median lethal dose assay to bifenazate from *T. urticae* collected in mint fields and hop yards from Washington.**

| Population | LD$_{50}$ (ppm) | 95% CI | Slope ± SEM | X$^2$ | df | RR* |
|---|---|---|---|---|---|---|
| Laboratory susceptible | 0.53 | 0.46–0.66 | 3.35±0.53 | 7.48 | 14 | 1 |
| 1—Hops | 1.23 | 0.42–2.01 | 1.19+0.27 | 14.92 | 14 | 2.22 |
| 2—Hops | 0.9 | 0.17–1.63 | 1.07+0.28 | 14.39 | 14 | 1.64 |
| 5—Hops | 2.17 | 1.35–3.16 | 1.53+0.28 | 15.84 | 14 | 3.94 |
| Laboratory susceptible | 1.3 | 0.47–2.12 | 1.34±0.28 | 27.3 | 14 | 1 |
| 11—Mint | 2.13 | 1.74–2.60 | 3.64±0.56 | 15.6 | 14 | 1.64 |
| 12—Mint | 9.51 | 4.68–92.48 | 0.91±0.23 | 21.1 | 14 | 4.46 |
| 13—Mint | 8.66 | 5.09–90.77 | 0.76±0.23 | 12.9 | 14 | 4.06 |

bifenazate (G126S) resistance was not successfully amplified in any of the samples. This technology allowed us to examine the relative fixation of genotypes within a sample population. In population 1 (collected from hops in Washington), we noted that 98.2% of the reads encoded for SNPs that are associated with etoxazole (I1017F) resistance, suggesting a highly fixed mutation within this population, while for population 9 (collected from mint in Idaho) we noted that 91.7% of the population had F1538I. This suggests that 8.3% of the reads were still wild-type and the population has not become homozygous for the resistance associated allele. This information provides additional insight into the relative frequency of the resistance associated genotypes within field collected populations which may be useful in choosing effective acaricides and integrated pest management approaches.

The results from the TaqMan qPCR and targeted-sequencing assays were compared for F1538I (Bifenthrin) and I1017F (Etoxazole) to confirm results (Table 2). The results from both methods had high agreement. The targeted-sequencing approach was more sensitive and provided accurate data on the proportion of the presence of the frequency of the allele. However, the TaqMan assay was able to provide accurate calls in 23 out of 24 of the runs. The TaqMan assay predicted population 2 (collected from hops in Washington) as a completely fixed population for I1017F when, in fact, there was a high proportion of wildtype alleles (17.45%). As such, the population should be considered heterozygous.

## Median lethal dose assay

Median lethal dose assays were conducted on 7 populations of mites: 3 collected from hops, 3 collected from mint in Washington, and a susceptible laboratory raised population (Table 3). The populations of mites collected from hops were all heterozygous for bifenazate resistance, while the populations collected from mint all had susceptible phenotypes. The results from the LD$_{50}$ experiment on field collected mites from hops and mint suggested that none of the examined populations had a resistant phenotype compared to the laboratory population. The results from the LD$_{50}$ assay suggest that even though the populations from hops had heterozygous resistant phenotypes, the expression of that allele does not accurately predict whether the population has a resistant phenotype. The populations collected in mint had a more resistant phenotype than the mites collected in hops when exposed to bifenazate. The results suggest that using this specific marker alone is not a significant predicator of resistance to bifenazate.

## Discussion

In this investigation we examined three mutations in field populations of *T. urticae* using two molecular diagnostic approaches. These mutations have been previously linked to acaricide resistance phenotypes and include the mutation G126S in cytochrome b, F1538I, in a voltage

gated sodium channel, and I1017F in the chitin synthase 1 gene, and have all been shown to confer varying levels of resistance to commonly used acaricides [19, 20, 23]. We examined these markers using two different diagnostic approaches: a pooled DNA TaqMan qPCR and targeted-sequencing. The TaqMan qPCR has been suggested as a molecular tool for growers to monitor resistance status of *T. urticae* [21, 22] and we set out to determine if a TaqMan-based approach would be a quick and reliable tool for growers to provide information regarding resistant phenotypes of local populations. Our findings confirm its utility to detect mutations in *T. urticae* and further demonstrates that it can be used for field-collected populations. Because it is not realistic to examine individual mite samples for an agriculturally relevant screening tool, we chose to examine pooled DNA samples for each field location in order to detect the presence of different genotypes within each population using a TaqMan-based approach. While this does not provide the allelic frequency, it may assist growers with determining appropriate integrated pest management decisions based on the presence of known mechanisms of resistance. We also confirmed the TaqMan results were accurate using the complementary (and, in some cases, more sensitive) targeted-sequencing approach. The targeted-sequencing provided insight into the relative proportion of allele frequencies that encodes for resistance within a population of *T. urticae* and allowed us to test if the TaqMan qPCR-based approach provides similar results. The findings of this investigation suggest that a TaqMan based approach can provide quick data from field collected population of *T. urticae*. However, how a grower would interpret the results in the context of pest management decisions needs to be further addressed. One suggestion is that the grower could make a decision regarding chemical application based on how fixed the genotype is in the field. Fields that are homozygous susceptible or homozygous resistant pose little concern for determining if a pesticide application should be made, as susceptible fields could be treated while the resistant fields should not be. The harder decisions come when fields are heterozygous. While future investigations are required in order to make this technique useful in those management decisions, the addition of molecular tools to current integrated pest management practices (including field scouting, insect traps, and current resistant detection methods) should be considered [11, 28, 29].

Using established primers and probes, a molecular laboratory can quickly assess the presence of SNPs that are associated with acaricide resistance using a TaqMan qPCR approach. This method includes sampling *T. urticae*, removing mite samples from leaves, extracting DNA, and running the TaqMan assay, which can all be done within roughly 4 hours depending on DNA extraction method. We noted that both DNA extraction methods provided suitable DNA for TaqMan PCR, however, the Zymo Research Quick-DNA Tissue/ Insect Microprep kit was quicker and easier to use. Understanding the results can be more challenging, including determining appropriate thresholds and actions for integrated pest management approaches. Determining resistance for a heterozygous population, where a portion of the population has the resistant allele and others do not, may prove challenging using the TaqMan assay alone. More data regarding the frequency of known resistance mutations may provide a more comprehensive picture of the resistance status of the population of *T. urticae*. For this, we examined the use of Illumina-based targeted-sequencing, which provided a reasonable estimate of the overall frequency of the mutation within a population. Using the same primers to detect both the wildtype and resistance associated alleles, we can deduce that the amplification frequency will be the same for each allele and the total number of reads for each allele can be used to estimate the frequency within a population. Both approaches are subject to error if there are unknown mutations in primer binding sites. Nonetheless, assessing the allele frequencies within a population sample may prove valuable to assist a grower in determining whether to apply a particular acaricide. While the targeted-sequencing is more sensitive for

assessing allele frequencies, it takes a significantly longer time to generate the data (amplifications, Illumina library prep and sequencing), which can be problematic for growers. The Taq-Man qPCR and targeted-sequencing, however, provided similar results, with 23 of the 24 of the pooled samples assessed agreeing between the two methods. While the TaqMan assay could not determine the allelic frequency, it can provide information regarding whether a population is wildtype, heterozygous, or a completely fixed mutant.

Genotypic data generated from the TaqMan qPCR and targeted-sequencing provides significant insight on resistance phenotypes of *T. urticae* populations and acaricide input within Idaho and Washington. A pyrethroid (bifenthrin) has never been registered for use on mint in the Pacific Northwest. The TaqMan qPCR and targeted-sequencing data for F1538I (Bifenthrin) suggested that most mint field populations of mites have some proportion of a resistance associated genotype for Bifenthrin. Bifenthrin is used commonly in silage corn [30] which is grown intensely around mint fields in Idaho and Washington. Throughout the growing season, corn fields are treated routinely with pyrethroids and *T. urticae* could parachute into the adjacent mint fields as the silage corn dries down in the fall. This movement between fields could explain why we see the markers that have been associated with resistance to bifenthrin in mites collected in mint, given the fact that no pyrethroids to date have been registered for use in mint fields. The data for G126S (Bifenazate) suggests that most populations of *T. urticae* from hops have a heterozygous phenotype, while mutations representing a resistant phenotype were rarely observed in *T. urticae* collected in mint in Idaho or Washington. Bifenazate is applied rarely to mint but heavily used in hops in the Pacific Northwest, which may explain the frequency of the resistant alleles found in this investigation. Etoxazole has cross-resistance with both hexythiozox and clofentazine. Washington mint growers have applied hexythiozox routinely since the late 1980s and the hop growers are often applying both hexythiozox and etoxazole, yearly, in the hop yards in Washington. The genotypic data for etoxazole matched prior expectations in Idaho and Washington, where we observed resistant genotypes in hops yards in Washington and no resistant genotypes in mint fields in Idaho where etoxazole is not commonly used.

To confirm the results of the genotypic assays, we performed an $LD_{50}$ assessment on bifenazate resistance. We decided to confirm resistance to one of the chemicals we explored, bifenazate, which can be predicted with the mutation G126S in the cytochrome b gene. The genotypic data suggested that *T. urticae* collected from hop yards and mint fields from Washington would demonstrate different resistant phenotypes. The $LD_{50}$ assessment for *T. urticae* collected from mint fields demonstrated a higher $LD_{50}$ value than *T. urticae* collected from hop yards. The results of the $LD_{50}$ study suggested the exact opposite of the genetic data, with populations collected from mint fields demonstrating slightly higher bifenazate resistance than hop yards. Recent studies agree with our findings and suggest that the G126S mutation may not be a good indicator that a population is resistant to bifenazate [24]. It is likely that multiple markers will be required to accurately detect resistance in field populations and further that the G126S mutation is not an optimal indicator of resistance status. The mutation F1538I, which has been previously associated with bifenthrin resistance, has also been suggested to be a poor predictor of resistance phenotypes [25]. In the future, all SNPs being considered for the detection of acaricide resistance using a TaqMan approach should be validated through $LD_{50}$ assessments for each specific acaricides. In this investigation we were only able to conduct an $LD_{50}$ assessment with bifenazate and, as such, further investigations should be conducted to confirm that the genotyping data matches resistance phenotypes for bifenthrin and etoxazole. Other mechanism of acaricides resistance (besides diagnostics SNPs) should also be considered when determining whether a population of mites has resistance phenotypes to acaracides. These mechanisms may include phase 1 and 2 detoxification and behavioral resistance [29,

31]. In this investigation the list of mutations explored was not extensive and there are mutations and mechanisms of resistance that are not detectable using TaqMan methodology [25, 29]. In the future, this approach could be used on many different insect species in which SNPs are associated with resistant phenotypes [32–34].

In this investigation we examined the use of using two different molecular approaches to detect resistance associated genotypes of *T. urticae* collected from mint and hops in the Pacific Northwest. The data suggests that TaqMan qPCR can be used to quickly genotype *T. urticae* collected from the field. However, the interpretation of the data might pose problems and concerns for integrated pest management decisions, including developing appropriate resistance allele frequency cutoffs for acaricide applications with populations that have a heterozygous genotype. We further demonstrated that the presence the G126S mutation may not accurately predict bifenazate resistance and should not be used to predict actual resistance profiles. While these genetic approaches are constantly evolving based on our knowledge of putative and nearby polymorphisms (crucial for a stable assay), further work is needed to validate the importance of specific mutations and the metrics used for determining overall resistance profiles within a population before they can be used to inform pest management decisions.

## Supporting information

**S1 Table. TaqMan primers and probes.**
(DOCX)

## Author Contributions

**Conceptualization:** Justin Clements.

**Data curation:** Silas Shumate, Maggie Haylett, Justin Clements.

**Formal analysis:** Silas Shumate, Maggie Haylett, Justin Clements.

**Funding acquisition:** Doug Walsh, Justin Clements.

**Investigation:** Silas Shumate, Maggie Haylett, Brenda Nelson, Nicole Young, Kurt Lamour, Doug Walsh, Justin Clements.

**Methodology:** Nicole Young, Kurt Lamour, Doug Walsh, Justin Clements.

**Project administration:** Justin Clements.

**Supervision:** Justin Clements.

**Visualization:** Benjamin Bradford.

**Writing – original draft:** Silas Shumate, Maggie Haylett, Brenda Nelson, Justin Clements.

**Writing – review & editing:** Silas Shumate, Maggie Haylett, Brenda Nelson, Nicole Young, Kurt Lamour, Doug Walsh, Benjamin Bradford, Justin Clements.

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
