## [Decision Letter · Decision Letter 0]

10 Aug 2022

PONE-D-22-09155Detection of bifenthrin, bifenazate, etoxazole resistance in Tetranychus urticae collected from peppermint fields and hop yards using targeted sequencing and TaqMan approaches in pooled DNA samplesPLOS ONE

Dear Dr. Justin Clements,

Thank you for submitting your manuscript to PLOS ONE. After careful consideration, we feel that it has merit but does not fully meet PLOS ONE’s publication criteria as it currently stands. Therefore, we invite you to submit a revised version of the manuscript that addresses the points raised during the review process.

We look forward to receiving your revised manuscript.

Kind regards,

Islam Hamim, PhD

Academic Editor

PLOS ONE

Journal Requirements:

This research was supported in part by funding from United States Department of Agriculture Specialty Crop Initiative 2021-51181-35901, awarded to DW and JC, a Mint Industry Research Council grant awarded to DW and JC, and a Washington Mint Commission grant awarded to DW and JC.

This research was supported in part by funding from United States Department of Agriculture Specialty Crop Initiative 2021-51181-35901, awarded to DW and JC, a Mint Industry Research Council grant awarded to DW and JC, and a Washington Mint Commission grant awarded to DW.

However, funding information should not appear in the Acknowledgments section or other areas of your manuscript. We will only publish funding information present in the Funding Statement section of the online submission form. 

This research was supported in part by funding from United States Department of Agriculture Specialty Crop Initiative 2021-51181-35901, awarded to DW and JC, a Mint Industry Research Council grant awarded to DW and JC, and a Washington Mint Commission grant awarded to DW and JC.

Additional Editor Comments:

Authors must respond to all reviewers' comments and make the necessary revisions to their manuscripts.

Reviewers' comments:

Reviewer's Responses to Questions

**Comments to the Author**

1. Is the manuscript technically sound, and do the data support the conclusions?

Reviewer #1: Yes

Reviewer #2: Yes

Reviewer #3: Yes

Reviewer #4: Yes

Reviewer #5: Partly

Reviewer #6: Partly

2. Has the statistical analysis been performed appropriately and rigorously? 

Reviewer #1: Yes

Reviewer #2: N/A

Reviewer #3: Yes

Reviewer #4: N/A

Reviewer #5: N/A

Reviewer #6: Yes

3. Have the authors made all data underlying the findings in their manuscript fully available?

Reviewer #1: Yes

Reviewer #2: Yes

Reviewer #3: Yes

Reviewer #4: Yes

Reviewer #5: Yes

Reviewer #6: Yes

4. Is the manuscript presented in an intelligible fashion and written in standard English?

Reviewer #1: Yes

Reviewer #2: Yes

Reviewer #3: Yes

Reviewer #4: No

Reviewer #5: Yes

Reviewer #6: Yes

5. Review Comments to the Author

Reviewer #1: Abstract

The abstract need to rewritten with key information.

Rewrite the statement “The results suggest the TaqMan approach accurately genotypes T. urticae populations collected from agricultural fields” for better understanding.

Introduction

The molecular mechanism for acaricide resistance may be improved from following latest references

1. https://doi.org/10.1016/j.pestbp.2021.104985

2. https://doi.org/10.11158/saa.26.8.10

3. https://doi.org/10.1186/s13071-020-04227-7

Materials & Method

1. Provide the reference for CethylTrimethyl Ammonium Bromide (CTAB) method and company details for Zymo Research Quick-DNA Tissue/132 Insect Microperp kit.

2. Line no 135, specify type and wavelength of UV light

3. DNA extraction procedure may be shortened by providing suitable reference and the modification made in protocol. Similarly, the kit method shortened by citing manufacturers’ instruction.

4. Line no 198 replicate or replication

Result & Discussion

1. Authors should explain the reasons for the samples did not amplify in Table 3 in results.

Reviewer #2: Dear Dr. Islam Hamim

Manuscript #: PONE-D-22-09155

Title: Detection of bifenthrin, bifenazate, etoxazole resistance in Tetranychus urticae collected from peppermint fields and hop yards using targeted sequencing and TaqMan approaches in pooled DNA samples

Whether the use of using two different molecular approaches to determine the resistance genotypes present in T. urticae from collected field are the main question in this study. For this aim, Authors examined the use of a TaqMan qPCR-based approach to determine acaricide resistance genotypes in field-collected populations of T. urticae from peppermint fields and hop yards in the Pacific Northwest of the United States and confirmed the results with a multiplex targeted sequencing. The results shown the TaqMan approach accurately genotypes T. urticae populations collected from agricultural fields. However, as the researchers of this method stated, it needs to be further developed for the correct determination of resistance in T. urticae. Therefore, this manuscript contains content of applicable value. This manuscript is appropriate to be published in the PLOS ONE after some corrections that I have mentioned below:

1. In "Running Title", acaricide should be used instead of miticide.

2. The abstract should be reviewed and some conclusions should be given.

3. Line 122 “dry” should be removed.

4. Information about the laboratory population of T. urticae should be given in the material and method.

5. In DNA isolation from Tetranychus urticae section, the mite stage used in DNA isolation should be specified.

6. In the determination of the bifenazate L50 value, 4 doses were used, except for the control. At least 5 doses were required to determine reliable LC50 (Yu,2015, Simon, J. Y. (2015). The toxicology and biochemistry of insecticides. CRC press.).

7. In the medium lethal dose assay section, how bifenazate was applied to mites should be written clearly.

8. Line 351, mint should be written instead of mites.

Reviewer #3: Detection of bifenthrin, bifenazate, etoxazole resistance in Tetranychus urticae collected from peppermint fields and hop yards using targeted sequencing and TaqMan approaches in pooled DNA samples

Authors Shumate et al. presented a study to detect SNPs that are associated with acaracide resistance in Tetranychus urticae by using targeted sequencing and TaqMan approaches. Their results showed that both diagnostic methods exhibited consistent results, but TaqMan approach may have potential to be used in the field since this method is relatively easier and taking less time compared to targeted sequencing. However, when authors interpreted the results of phenotypes and genotypes, more discussion is needed. G126S is a mutation associated with low level of bifenezate resistance (several citations needed here) but recent study (reference 24) suggested that G126S may not link to bifenezate resistance. Besides target site insensitivity (mutations on the target proteins that have been functional studied), other mechanisms such as enhanced metabolic detoxification (P450s, esterases, GSTs) or sequestrations (ABC transporters) may also play roles in acaricide resistance. Authors only cited 24 papers in their manuscript. More previous studies associated with this topic should be discussed when authors interpreted their results and pointed out future directions.

1. Change to a more specific title: Using targeted sequencing and TaqMan approaches to detect acaricide resistance associated SNPs in Tetranychus urticae collected from peppermint fields and hop yards

2. Material and Method

Page 11, line 115: detailed information of these 13 field-collected populations (location-latitude and longitude, collection time) and the lab susceptible population should be listed in the main text or in a supplementary table.

3. Material and Method

Medium lethal dose assay: not very clear. Some details missed, such as which method was used for pesticide treatment? Any machine used for bioassay? Please cite some previous studies/publications or provide more details here.

4. Table 1 and Table 2 are same as the Table 3. Suggested to delete Table 1&2 and only leave the Table 3

5. Table 1 and Table 3: Population 1&2 in I1017F (Tagman) shows “Resistance”, however, other populations showed “wildtype” or “heterozygous”. Suggest changing “Resistance” to “heterozygous” or “homozygous”

6. Table 4. The bioassay data only exhibited resistance to bifenezate but did not include other two acaricides. Since the purpose of this experiment is to show if phenotype and genotype matches. G126S is not a good predictor for bifenezate resistance, then other two bioassay data should be more important. Authors should add data or at least discuss other possible reasons (previous bioassay data, other resistance mechanisms besides target site insensitivity, etc).

Reviewer #4: Review report PONE-D-22-09155: Detection of bifenthrin, bifenazate, etoxazole resistance in Tetranychus urticae collected from peppermint fields and hop yards using targeted sequencing and TaqMan approaches in pooled DNA samples

Tetranychus urticae is an important pest and recorded on more than 1,000 different plant species infested worldwide. Although several methods have been used frequently, farmers mostly rely on chemical pesticides against T. urticae. But the acaricide resistance can develop through multiple mechanisms, including enhanced metabolic breakdown of acaricides, target site insensitivity, and behavioral resistance. The authors examined the use of a TaqMan qPCR-based approach to determine acaricide resistance genotypes in field-collected populations of T. urticae from peppermint fields and hop yards in the Pacific Northwest of the United States and confirmed the results with a multiplex targeted sequencing. Although the MS is well written, the following points need to be addressed.

L 26-27: Multiple acaricides are currently registered for control including bifenazate, bifenthrin, and extoxazole > Multiple acaricides including bifenazate, bifenthrin, and extoxazole are currently registered for control T. urticae.

L 29-30: Not clear. What is integrated pest management tools?

L 31-34: The authors mentioned “The detection of these mutations through TaqMan qPCR has been suggested as a practical, quick, and reliable tool to inform agricultural producers of acaricide resistance phenotypes present within their fields and have potential utility for making appropriate acaricide application and integrated pest management decisions”—Here is the major concern that the authors emphasis the TaqMan qPCR throughout the MS to determine acaricide resistance genotypes. This is not a new method to determine to detect different chemical resistance status of spider mites. It would be better to compare with other method and confirm the accuracy to determine resistance genotypes against specific chemicals.

L 71-72: Any reference(s).

L 77-79: Pls rephrase

L 197: Why the author performs the median lethal dose assay only for bifenazate?

L 277-280: Add reference(s)

L 281-284: If the method is already well-known molecular tool for growers to detect different chemical resistance status of T. urticae, what is the novelty of present research. (Pls check reference 19, 20)

None of the scientific name is Italic in reference section.

Reviewer #5: Clements et al. used TaqMan qPCR/Multiplex targeted sequencing to determine acaricide resistance genotypes in field populations of Tetranychus urticae in the Northwest of the USA. Their results indicate that TaqMan approach accurately genotypes T. urticae populations but according to the authors interpretation of results is challenging.

The difficult interpretation of the results might partly be caused by the fact that G126S in cytb (used by the the authors for bifenazate resistance screening) is no longer considered as a bifenazate resistance mutation (Xue et al. 2021, PMS). The authors do acknowledge Xue et al. 2021 but only in the penultimate paragraph of their manuscript. I think the authors should be more upfront about this, and change abstract/introduction accordingly (i.e. that their results confirm findings of Xue et al. 2021). They could change title to "Detection of acaricide resistance in Tetranychus urticae collected from ....", mention in the abstract that "G126S is not a good bifenzate resistance indicator" ; also rephrase line 88: "This mutation has been shown to confer resistance" and line 107: "G126S is a well-defined marker of resistance", and change line 211 to "TaqMan quantitive PCR detection of target-site mutations". Wihthout these changes, I believe the manuscript will contribute to the misinterpretation of the role of the G126S in bifenazate resistance and cannot be accepted for publication.

Other comments:

line 90: I1017F in Chs1 was also shown to confer resistance against hexythiazox and clofentezine (10.1016/j.ibmb.2014.05.004). Authors should include this reference in the introduction

DNA extraction methods: The authors do mention two extraction methods (Zymo kit vs CTAB) but do not compare results between the two methods (e.g. 260/280 values or yield?). Author should include this comparison in the main text of the manuscript.

Table 3 shows significant overlap with Table 1/2. Authors should combine these three tables into one table.

line 279: I1017F mutation has also been screened in T. urticae populations using resitriction digest/CT method (j.pestbp.2017.04.003); authors should integrate this reference in the Discussion or Introduction

line 321: other VGSC mutations have also been implicated in bifenthrin resistance (L1024V) and non target site

mediated resistance mechanisms are also involved in bifenthrin resistance (see Riga et al. 2014, 10.1038/s41598-017-09054-y), which cannot be detected by the Taqman assay developed in this study. Author should integrate this reference in the Discussion section.

Reviewer #6: The manuscript “Detection of bifenthrin, bifenazate, etoxazole resistance in Tetranychus urticae collected from peppermint fields and hop yards using targeted sequencing and

TaqMan approaches in pooled DNA samples” by Shumate et al., describes the analysis of 13 T. urticae populations for the bifenthrin, bifenazate, etoxazole resistance. They used two methods to assess target site mutations that were reported to be associated with resistance to these pesticides, TaqMan and Multiplex targeted-sequencing (not for bifenazate), and have determined the actual resistance status of collected field mite populations. They checked the correlation/predictability between these parameters and found that target SNPs were congruently detected with either of the sequencing methods, but that genotyping information did not correlate with the actual resistance status of tested populations, indicating that tested polymorphisms are not tightly correlated with the resistance in T. urticae. Authors confirm the utility of TaqMan to detect mutations in T. urticae and further demonstrate that it can be used for field-collected populations.

The information presented is useful and is adding to several attempts to develop diagnostic tools to predict pesticide resistance in mite populations based on genotype information. The main challenges there is not necessarily detection of the allelic frequency, but their correlation with the resistance phenotype. Next, what are cut offs for defining a population as resistant vs susceptible based on allele frequency, even if it is 100% predictive of the resistance. Manuscript would benefit from additional data, explanations and some corrections.

a) Line 221: Why did authors choose bifenazate (G126S), see Xue W, Wybouw N, Van Leeuwen T. The G126S substitution in mitochondrially encoded cytochrome b does not confer bifenazate resistance in the spider mite Tetranychus urticae. Experimental and Applied Acarology. 2021 Dec;85(2):161-72. At least they should detect S141F SNP as well, as per DOI:10.1038/s41598-017-09054-y.

Likewise, what was the rationale to detect VGSC (F1538I) when it was reported to be a poor predictor of mite resistance to bifenthrin?

chs (I1017F) should be tightly correlated to etoxazole resistance.

b) Line 250: “The TaqMan assay predicted population 2 (collected from hops in Washington) as a completely resistant population when, in fact, there was a high proportion of wildtype alleles (17.45%). As such, the population should be considered heterozygous.” Authors refer to Etoxazole data? They should clearly state it. In addition, they should define criteria for calling populations heterozygous/homozygous, and in particular resistant vs sensitive. Population 2 should give rise to 68% of mites homozygous to I1017F SNP and thus predicted to be resistant. Is this population resistant or susceptible according to authors? What should be cut offs and how are they defined?

c) Line 259:

Median lethal dose assay was performed only for bifenazate resistance. As per Xue W, Wybouw N, Van Leeuwen T. The G126S substitution in mitochondrially encoded cytochrome b does not confer bifenazate resistance in the spider mite Tetranychus urticae. Experimental and Applied Acarology. 2021 Dec;85(2):161-72., this is the least predictable association. Authors concur in line 269, but they should have known it ahead of the time.

Authors should add etoxazole bioassays.

d) Line 284-297: “Our findings confirm the utility of TaqMan to detect mutations in T. urticae and further demonstrates that it can be used for field-collected populations.” “However,

how a grower would interpret the results in the context of pest management decisions needs to be

further addressed.” Authors have to define or at least propose how TaqMan data can be used for diagnostic decision.

e) Line 321: “A pyrethroid (bifenthrin) has never been registered for use on mint in the Pacific Northwest. The TaqMan qPCR and targeted-sequencing data for F1538I (Bifenthrin) suggested that most mint field populations of mites have some proportion of a resistance phenotype for Bifenthrin.”

Careful!!! There was no resistance bioassay performed. According to DOI:10.1038/s41598-017-09054-y, this SNP is not very predictive of the resistance state.

f) Line 339: “To confirm the results of the genotypic assays, we performed an LD50 assessment on bifenazate resistance. We decided to confirm resistance to one of the chemicals we explored, bifenazate, which can be predicted with the mutation G126S in the cytochrome b gene.” This contradicts data presented and data in DOI:10.1038/s41598-017-09054-y and Xue et al., 2021.

Corrections:

Grammar in sentences in lines: 86-88, 99-100

6. PLOS authors have the option to publish the peer review history of their article (what does this mean?). If published, this will include your full peer review and any attached files.

Reviewer #1: **Yes: **Devendra Jain

Reviewer #2: No

Reviewer #3: No

Reviewer #4: No

Reviewer #5: No

Reviewer #6: **Yes: **Vojislava Grbic

---

## [Author Response · Author response to Decision Letter 0]

15 Sep 2022

Reviewer #1

Abstract

The abstract need to rewritten with key information.

Rewrite the statement “The results suggest the TaqMan approach accurately genotypes T. urticae populations collected from agricultural fields” for better understanding.

We have added additional information into the abstract regarding key findings and have reworded the above sentence.

Introduction

The molecular mechanism for acaricide resistance may be improved from following latest references

1. https://doi.org/10.1016/j.pestbp.2021.104985

2. https://doi.org/10.11158/saa.26.8.10

3. https://doi.org/10.1186/s13071-020-04227-7

We have added two of the above citations to improve the manuscript. We refrained from using the citation from Cossío-Bayúgar, as it was focused on Rhipicephalus microplus (tick parasite).

Materials & Method

1. Provide the reference for CethylTrimethyl Ammonium Bromide (CTAB) method and company details for Zymo Research Quick-DNA Tissue/132 Insect Microperp kit.

We have added the company name for both the CTAB reagent and Zymo Kit.

2. Line no 135, specify type and wavelength of UV light

We have added the wavelength, which was 254 nm

3. DNA extraction procedure may be shortened by providing suitable reference and the modification made in protocol. Similarly, the kit method shortened by citing manufacturers’ instruction.

In the methods we have added the company name and that the CTAB reaction was a modified reaction (which is the reason why we have included a detailed procedure).

4. Line no 198 replicate or replication

Replicate is the correct term

Result & Discussion

1. Authors should explain the reasons for the samples did not amplify in Table 3 in results.

Sample 12 did not amplify in the Illumina reaction. We are unaware of why this sample was not able to be amplified, as the reaction was sent to a third party for amplification. Multiple reasons could be behind the non-amplification on the Illumina run. Because this amplification was conducted by a third party, we do not want to speculate on the reasoning behind why sample 12 did not amplify.

Reviewer #2 

Dear Dr. Islam Hamim

Manuscript #: PONE-D-22-09155

Title: Detection of bifenthrin, bifenazate, etoxazole resistance in Tetranychus urticae collected from peppermint fields and hop yards using targeted sequencing and TaqMan approaches in pooled DNA samples

Whether the use of using two different molecular approaches to determine the resistance genotypes present in T. urticae from collected field are the main question in this study. For this aim, Authors examined the use of a TaqMan qPCR-based approach to determine acaricide resistance genotypes in field-collected populations of T. urticae from peppermint fields and hop yards in the Pacific Northwest of the United States and confirmed the results with a multiplex targeted sequencing. The results shown the TaqMan approach accurately genotypes T. urticae populations collected from agricultural fields. However, as the researchers of this method stated, it needs to be further developed for the correct determination of resistance in T. urticae. Therefore, this manuscript contains content of applicable value. This manuscript is appropriate to be published in the PLOS ONE after some corrections that I have mentioned below:

1. In "Running Title", acaricide should be used instead of miticide.

Changed as suggested

2. The abstract should be reviewed and some conclusions should be given.

We have added conclusion to the abstract as suggested

3. Line 122 “dry” should be removed.

Removed as suggested

4. Information about the laboratory population of T. urticae should be given in the material and method.

We have added, “A laboratory population of T. urticae was used as a baseline reference for the LD50 assays. This population represents a historically susceptible population that has been housed at the IAREC center in Processer Washington for multiple generations, has not been subjected to any chemical exposure over those generations, and has been raised on beans.” To describe the susceptible laboratory population

5. In DNA isolation from Tetranychus urticae section, the mite stage used in DNA isolation should be specified.

 We have added, “DNA extractions were conducted on pooled motile mites encompassing multiple life stages.”

6. In the determination of the bifenazate L50 value, 4 doses were used, except for the control. At least 5 doses were required to determine reliable LC50 (Yu,2015, Simon, J. Y. (2015). The toxicology and biochemistry of insecticides. CRC press.).

Yu (2015) suggests, “The animals available for testing are divided at random into several groups, usually six. One of these is used as the control group…” We agree that more doses would refine the estimate but there is no defined number of doses required for LD50 assessment. Many studies have utilized less than 6 doses for LD50 studies and the tested rates were the available rate at the time of the assay. 

7. In the medium lethal dose assay section, how bifenazate was applied to mites should be written clearly.

We have added that treatments were sprayed onto the leaf discs using a potter spray tower in the methods section

8. Line 351, mint should be written instead of mites.

We have changed mites to mint as suggested. Thank you for the correction.

Reviewer #3: 

Detection of bifenthrin, bifenazate, etoxazole resistance in Tetranychus urticae collected from peppermint fields and hop yards using targeted sequencing and TaqMan approaches in pooled DNA samples

Authors Shumate et al. presented a study to detect SNPs that are associated with acaracide resistance in Tetranychus urticae by using targeted sequencing and TaqMan approaches. Their results showed that both diagnostic methods exhibited consistent results, but TaqMan approach may have potential to be used in the field since this method is relatively easier and taking less time compared to targeted sequencing. However, when authors interpreted the results of phenotypes and genotypes, more discussion is needed. G126S is a mutation associated with low level of bifenezate resistance (several citations needed here) but recent study (reference 24) suggested that G126S may not link to bifenezate resistance. Besides target site insensitivity (mutations on the target proteins that have been functional studied), other mechanisms such as enhanced metabolic detoxification (P450s, esterases, GSTs) or sequestrations (ABC transporters) may also play roles in acaricide resistance. Authors only cited 24 papers in their manuscript. More previous studies associated with this topic should be discussed when authors interpreted their results and pointed out future directions.

We have added more citation as requested by reviewers throughout the manuscript and further provide more interpretation of the results.

1. Change to a more specific title: Using targeted sequencing and TaqMan approaches to detect acaricide resistance associated SNPs in Tetranychus urticae collected from peppermint fields and hop yards

Changed as suggested

2. Material and Method

Page 11, line 115: detailed information of these 13 field-collected populations (location-latitude and longitude, collection time) and the lab susceptible population should be listed in the main text or in a supplementary table.

We have provided more information about the lab susceptible population within the methods. We are unable to provide the exact field location due to confidentiality agreements with the growers that we work with.

3. Material and Method

Medium lethal dose assay: not very clear. Some details missed, such as which method was used for pesticide treatment? Any machine used for bioassay? Please cite some previous studies/publications or provide more details here.

We have added more information about the method of treating the leaf disks. “Treatments were sprayed onto the leaf surface with a potter spray tower. Ten gravid adult female T. urticae were placed on each leaf disk. Each bioassay arena for each of the tested rates was treated with 2 ml of the dilute solution. After 24 hours mites, were evaluated for mortality.”

4. Table 1 and Table 2 are same as the Table 3. Suggested to delete Table 1&2 and only leave the Table 3

We have removed table 2, as it has the same data incorporated into table 3. However, Table 1 has unique and important information.

5. Table 1 and Table 3: Population 1&2 in I1017F (Tagman) shows “Resistance”, however, other populations showed “wildtype” or “heterozygous”. Suggest changing “Resistance” to “heterozygous” or “homozygous”

We agree that this is confusing and have added a foot note to both tables. The reason why we do not list the resistance population as homozygous is that homozygous could refer to either wildtype or the resistant population, depending on the genotype.

6. Table 4. The bioassay data only exhibited resistance to bifenezate but did not include other two acaricides. Since the purpose of this experiment is to show if phenotype and genotype matches. G126S is not a good predictor for bifenezate resistance, then other two bioassay data should be more important. Authors should add data or at least discuss other possible reasons (previous bioassay data, other resistance mechanisms besides target site insensitivity, etc).

We agree that the additional bioassays would have been beneficial to this manuscript. Unfortunately, the data was not generated for the other two acaricide and cannot be reported on. We have added a section into the discussion regarding the other modes of detoxification could be responsible for miticide resistance as suggested by the reviewer. 

Reviewer #4: 

Review report PONE-D-22-09155: Detection of bifenthrin, bifenazate, etoxazole resistance in Tetranychus urticae collected from peppermint fields and hop yards using targeted sequencing and TaqMan approaches in pooled DNA samples

Tetranychus urticae is an important pest and recorded on more than 1,000 different plant species infested worldwide. Although several methods have been used frequently, farmers mostly rely on chemical pesticides against T. urticae. But the acaricide resistance can develop through multiple mechanisms, including enhanced metabolic breakdown of acaricides, target site insensitivity, and behavioral resistance. The authors examined the use of a TaqMan qPCR-based approach to determine acaricide resistance genotypes in field-collected populations of T. urticae from peppermint fields and hop yards in the Pacific Northwest of the United States and confirmed the results with a multiplex targeted sequencing. Although the MS is well written, the following points need to be addressed.

L 26-27: Multiple acaricides are currently registered for control including bifenazate, bifenthrin, and extoxazole > Multiple acaricides including bifenazate, bifenthrin, and extoxazole are currently registered for control T. urticae.

Changed.

L 29-30: Not clear. What is integrated pest management tools?

Removed sentence to avoid any confusion

L 31-34: The authors mentioned “The detection of these mutations through TaqMan qPCR has been suggested as a practical, quick, and reliable tool to inform agricultural producers of acaricide resistance phenotypes present within their fields and have potential utility for making appropriate acaricide application and integrated pest management decisions”—Here is the major concern that the authors emphasis the TaqMan qPCR throughout the MS to determine acaricide resistance genotypes. This is not a new method to determine to detect different chemical resistance status of spider mites. It would be better to compare with other method and confirm the accuracy to determine resistance genotypes against specific chemicals.

We agree that this method has been suggested as a tool previously. The authors wanted to make sure that this was clear. However, it has not been vetted in agriculturally relevant fields in a manner that would be practical to hops and mint growers. In this investigation, we worked with growers to sample fields for mites and processed the mites in a similar method (pooled samples) that a grower would encounter. This investigation was meant as a first step to further investigate TaqMan approaches to genotyping field populations of mites and additional studies should conducted to compare classical methods of resistance detection to TaqMan qPCR.

L 71-72: Any reference(s).

We have added a reference to support this section.

L 77-79: Pls rephrase

We have rephrased, “These polymorphisms are non-synonymous and may be the result of natural selection when they result in beneficial traits (16).” 

L 197: Why the author performs the median lethal dose assay only for bifenazate?

We agree that the additional bioassays would have been beneficial to this manuscript. Unfortunately, the data was not generated for the other two acaricide and cannot be reported on. Bifenazate at the time was the only acaricide for which we examined LD50 and we are no longer able to perform these studies.

L 277-280: Add reference(s)

We have added references for this section

L 281-284: If the method is already well-known molecular tool for growers to detect different chemical resistance status of T. urticae, what is the novelty of present research. (Pls check reference 19, 20)

In line 281 we suggested, “The TaqMan qPCR has been suggested as a molecular tool for growers to monitor resistance status of T. urticae (21,22) and we set out to determine if a TaqMan-based approach would be a quick and reliable tool for growers to provide information regarding resistant phenotypes of local populations.” While it has been suggested for this purpose, it has not been field tested or examined thoroughly.

None of the scientific name is Italic in reference section.

Changed as suggested

Reviewer #5: 

Clements et al. used TaqMan qPCR/Multiplex targeted sequencing to determine acaricide resistance genotypes in field populations of Tetranychus urticae in the Northwest of the USA. Their results indicate that TaqMan approach accurately genotypes T. urticae populations but according to the authors interpretation of results is challenging.

The difficult interpretation of the results might partly be caused by the fact that G126S in cytb (used by the the authors for bifenazate resistance screening) is no longer considered as a bifenazate resistance mutation (Xue et al. 2021, PMS). The authors do acknowledge Xue et al. 2021 but only in the penultimate paragraph of their manuscript. I think the authors should be more upfront about this, and change abstract/introduction accordingly (i.e. that their results confirm findings of Xue et al. 2021). They could change title to "Detection of acaricide resistance in Tetranychus urticae collected from ....", mention in the abstract that "G126S is not a good bifenzate resistance indicator" ; also rephrase line 88: "This mutation has been shown to confer resistance" and line 107: "G126S is a well-defined marker of resistance", and change line 211 to "TaqMan quantitive PCR detection of target-site mutations". Wihthout these changes, I believe the manuscript will contribute to the misinterpretation of the role of the G126S in bifenazate resistance and cannot be accepted for publication.

We agree with the reviewer and have changed the title of the manuscript. We mention that G126S is not a good indicator of resistance in the introduction and abstract. We feel that findings of this manuscript support the work that was done by Xue et al. 2021. Much of our work was conducted significantly before Xue et al. manuscript was published and feel how it is written is appropriate manner and supports their findings.

Other comments:

line 90: I1017F in Chs1 was also shown to confer resistance against hexythiazox and clofentezine (10.1016/j.ibmb.2014.05.004). Authors should include this reference in the introduction

Added as suggested

DNA extraction methods: The authors do mention two extraction methods (Zymo kit vs CTAB) but do not compare results between the two methods (e.g. 260/280 values or yield?). Author should include this comparison in the main text of the manuscript.

We have added additional information comparing the two extraction methods in the results as suggested by the reviewer.

Table 3 shows significant overlap with Table 1/2. Authors should combine these three tables into one table.

We have removed table 2, as it has much of the same information as table 3. We have left table 1, as it has unique and necessary information.

line 279: I1017F mutation has also been screened in T. urticae populations using resitriction digest/CT method (j.pestbp.2017.04.003); authors should integrate this reference in the Discussion or Introduction

Added as suggested

line 321: other VGSC mutations have also been implicated in bifenthrin resistance (L1024V) and non target site

mediated resistance mechanisms are also involved in bifenthrin resistance (see Riga et al. 2014, 10.1038/s41598-017-09054-y), which cannot be detected by the Taqman assay developed in this study. Author should integrate this reference in the Discussion section.

Added as suggested in the discussion.

Reviewer #6: 

The manuscript “Detection of bifenthrin, bifenazate, etoxazole resistance in Tetranychus urticae collected from peppermint fields and hop yards using targeted sequencing and

TaqMan approaches in pooled DNA samples” by Shumate et al., describes the analysis of 13 T. urticae populations for the bifenthrin, bifenazate, etoxazole resistance. They used two methods to assess target site mutations that were reported to be associated with resistance to these pesticides, TaqMan and Multiplex targeted-sequencing (not for bifenazate), and have determined the actual resistance status of collected field mite populations. They checked the correlation/predictability between these parameters and found that target SNPs were congruently detected with either of the sequencing methods, but that genotyping information did not correlate with the actual resistance status of tested populations, indicating that tested polymorphisms are not tightly correlated with the resistance in T. urticae. Authors confirm the utility of TaqMan to detect mutations in T. urticae and further demonstrate that it can be used for field-collected populations.

The information presented is useful and is adding to several attempts to develop diagnostic tools to predict pesticide resistance in mite populations based on genotype information. The main challenges there is not necessarily detection of the allelic frequency, but their correlation with the resistance phenotype. Next, what are cut offs for defining a population as resistant vs susceptible based on allele frequency, even if it is 100% predictive of the resistance. Manuscript would benefit from additional data, explanations and some corrections.

We agree with the reviewer that cutoff currently is not well defined and would be hard for a grower to determine when to consider a population as resistant. This is one of the fundamental reasons why we composed this manuscript to demonstrate this problem. “However, how a grower would interpret the results in the context of pest management decisions needs to be further addressed.”

a) Line 221: Why did authors choose bifenazate (G126S), see Xue W, Wybouw N, Van Leeuwen T. The G126S substitution in mitochondrially encoded cytochrome b does not confer bifenazate resistance in the spider mite Tetranychus urticae. Experimental and Applied Acarology. 2021 Dec;85(2):161-72. At least they should detect S141F SNP as well, as per DOI:10.1038/s41598-017-09054-y.

We agree with the reviewer that G126S is not a good indicator of resistance. We have addressed this in-depth throughout the manuscript. We feel that findings of this manuscript support the work that was done by Xue et al. 2021. All of our work was conducted before Xue et al. manuscript was published and feel how it is written is appropriate and supports their findings.

Likewise, what was the rationale to detect VGSC (F1538I) when it was reported to be a poor predictor of mite resistance to bifenthrin?

We used the F1538I gene due to the extensive research that revolve around this target candite. The targets that we choose were limited. In order for TaqMan qPCR to be a practical tool for growers, all known targets will have to have primer and probes designed for them.

chs (I1017F) should be tightly correlated to etoxazole resistance.

b) Line 250: “The TaqMan assay predicted population 2 (collected from hops in Washington) as a completely resistant population when, in fact, there was a high proportion of wildtype alleles (17.45%). As such, the population should be considered heterozygous.” Authors refer to Etoxazole data? They should clearly state it. In addition, they should define criteria for calling populations heterozygous/homozygous, and in particular resistant vs sensitive. Population 2 should give rise to 68% of mites homozygous to I1017F SNP and thus predicted to be resistant. Is this population resistant or susceptible according to authors? What should be cut offs and how are they defined?

The reviewer brings up an important point of cutoffs for wildtype and resistance phenotypes. The authors used Bio-Rad CFX Maestro software to classify TaqMan based results into homozygous wildtype, heterozygous, and homozygous resistance. We have clarified this in the methods. We confirmed the results of the TaqMan assay with a Multiplex PCR. The cutoff that we used to determine resistance using the Multiplex reaction was that wildtype or resistant population has to have at least 90% of the reads to be confirmed. We noted that this information was not included in the manuscript and have now included it.

c) Line 259:

Median lethal dose assay was performed only for bifenazate resistance. As per Xue W, Wybouw N, Van Leeuwen T. The G126S substitution in mitochondrially encoded cytochrome b does not confer bifenazate resistance in the spider mite Tetranychus urticae. Experimental and Applied Acarology. 2021 Dec;85(2):161-72., this is the least predictable association. Authors concur in line 269, but they should have known it ahead of the time.

Xue et al. Manuscript was published in December of 2021. The data for this manuscript was collected and processed before that manuscript was published. The findings of this manuscript further support their results.

Authors should add etoxazole bioassays.

We agree that additional bioassays would be beneficial to this study. However, at this time we are unable to perform additional assays (please refer to our cover letter). We believe the findings of the investigation are still relevant.

d) Line 284-297: “Our findings confirm the utility of TaqMan to detect mutations in T. urticae and further demonstrates that it can be used for field-collected populations.” “However,

how a grower would interpret the results in the context of pest management decisions needs to be further addressed.” Authors have to define or at least propose how TaqMan data can be used for diagnostic decision.

To discuss diagnostic decisions we have added, “However, how a grower would interpret the results in the context of pest management decisions needs to be further addressed. However, how a grower would interpret the results in the context of pest management decisions needs to be further addressed. One suggestion is that the grower could make a decision regarding chemical application based on how fixed the genotype is in the field. Fields that are homozygous susceptible or homozygous resistant pose little concern for determining if a pesticide application should be made, as susceptible fields could be treated while the resistant fields should not be. The harder decisions come when fields are heterozygous. Future investigations are required in order to make this technique useful in those management decisions.”

e) Line 321: “A pyrethroid (bifenthrin) has never been registered for use on mint in the Pacific Northwest. The TaqMan qPCR and targeted-sequencing data for F1538I (Bifenthrin) suggested that most mint field populations of mites have some proportion of a resistance phenotype for Bifenthrin.”

Careful!!! There was no resistance bioassay performed. According to DOI:10.1038/s41598-017-09054-y, this SNP is not very predictive of the resistance state.

Thank you for the correction, we changed phenotype to genotype.

f) Line 339: “To confirm the results of the genotypic assays, we performed an LD50 assessment on bifenazate resistance. We decided to confirm resistance to one of the chemicals we explored, bifenazate, which can be predicted with the mutation G126S in the cytochrome b gene.” This contradicts data presented and data in DOI:10.1038/s41598-017-09054-y and Xue et al., 2021.

We go into significant detail regarding that G126S is not a good marker for resistance. This is mentioned throughout the manuscript including, “To confirm the results of the genotypic assays, we performed an LD50 assessment on bifenazate resistance. We decided to confirm resistance to one of the chemicals we explored, bifenazate, which can be predicted with the mutation G126S in the cytochrome b gene. The genotypic data suggested that T. urticae collected from hop yards and mint fields from Washington would demonstrate different resistant phenotypes. The LD50 assessment for T. urticae collected from mint fields demonstrated a higher LD50 value than T. urticae collected from hop yards. The results of the LD50 study suggested the exact opposite of the genetic data, with populations collected from mint fields demonstrating slightly higher bifenazate resistance than hop yards. Recent studies agree with our findings and suggest that the G126S mutation may not be a good indicator that a population is resistant to bifenazate (24). It is likely that multiple markers will be required to accurately detect resistance in field populations and further that the G126S mutation is not an optimal indicator of resistance status.”

Corrections:

Grammar in sentences in lines: 86-88, 99-100

We have reworded these lines.

---

## [Decision Letter · Decision Letter 1]

8 Nov 2022

PONE-D-22-09155R1Using targeted sequencing and TaqMan approaches to detect acaricide resistance associated SNPs in Tetranychus urticae collected from peppermint fields and hop yardsPLOS ONE

Dear Dr. Justin Clements,

Thank you for submitting your manuscript to PLOS ONE. After careful consideration, we feel that it has merit but does not fully meet PLOS ONE’s publication criteria as it currently stands. Therefore, we invite you to submit a revised version of the manuscript that addresses the points raised during the review process. Authors need to address comments from Reviewer 4 and reviewer 6.

We look forward to receiving your revised manuscript.

Kind regards,

Islam Hamim, PhD

Academic Editor

PLOS ONE

Journal Requirements:

Additional Editor Comments:

Authors need to address comments from Reviewer 4 and reviewer 6.

Reviewers' comments:

Reviewer's Responses to Questions

**Comments to the Author**

1. If the authors have adequately addressed your comments raised in a previous round of review and you feel that this manuscript is now acceptable for publication, you may indicate that here to bypass the “Comments to the Author” section, enter your conflict of interest statement in the “Confidential to Editor” section, and submit your "Accept" recommendation.

Reviewer #2: All comments have been addressed

Reviewer #3: All comments have been addressed

Reviewer #4: (No Response)

Reviewer #5: All comments have been addressed

Reviewer #6: (No Response)

2. Is the manuscript technically sound, and do the data support the conclusions?

Reviewer #2: Yes

Reviewer #3: Yes

Reviewer #4: Partly

Reviewer #5: Yes

Reviewer #6: Partly

3. Has the statistical analysis been performed appropriately and rigorously? 

Reviewer #2: Yes

Reviewer #3: Yes

Reviewer #4: N/A

Reviewer #5: N/A

Reviewer #6: Yes

4. Have the authors made all data underlying the findings in their manuscript fully available?

Reviewer #2: (No Response)

Reviewer #3: Yes

Reviewer #4: Yes

Reviewer #5: Yes

Reviewer #6: Yes

5. Is the manuscript presented in an intelligible fashion and written in standard English?

Reviewer #2: Yes

Reviewer #3: Yes

Reviewer #4: Yes

Reviewer #5: Yes

Reviewer #6: Yes

6. Review Comments to the Author

Reviewer #2: Dear Dr. Islam Hamim

The authors have made the requested changes in MS and added new information. This manuscript is appropriate to be published in the PLOS ONE.

Reviewer #3: (No Response)

Reviewer #4: Review report PONE-D-22-09155R1: Detection of bifenthrin, bifenazate, etoxazole resistance in Tetranychus urticae collected from peppermint fields and hop yards using targeted sequencing and TaqMan approaches in pooled DNA samples

The authors examined the use of a TaqMan qPCR-based approach to determine acaricide resistance genotypes in field-collected populations of T. urticae from peppermint fields and hop yards in the Pacific Northwest of the United States and confirmed the results with a multiplex targeted sequencing. Although the MS is improved much, the following point need to be addressed.

The authors changed the title as ‘Using targeted sequencing and TaqMan approaches to detect acaricide resistance associated SNPs in Tetranychus urticae collected from peppermint fields and hop yards’

I appreciate their decision and would suggest specifying the acaricide name(s) in the title rather acaricide. Still, I am not convinced that without examining the median lethal dose assay of bifenthrin and etoxazole to detect the acaricide resistance associated SNPs. If this is not possible to perform these studies in this stage, that should be addressed in the discussion.

Reviewer #5: (No Response)

Reviewer #6: I find that revisions did not address the main problem of this manuscript. Yes, authors demonstrated that TaqMan protocol can identify SNPs in mite populations, BUT, authors do not state clearly that we do not have genetic markers for mite pesticide resistance (except etoxazole based on the prior research). Instead, the revised manuscript states in the abstract, line 33: “Within this investigation we examined the use of a TaqMan

34 qPCR-based approach to determine acaricide resistance genotypes in field-collected populations of T.

35 urticae from peppermint fields and hop yards in the Pacific Northwest of the United States and

36 confirmed the results with a multiplex targeted sequencing.” This is MISLEADING! Authors detected SNPs (genotypes), but not acaricide resistance.

Furthermore, in Introduction authors state, Line 81: ” SNPs in multiple, different

82 genes have become associated with the development of resistance to different acaricide chemistries

83 including bifenthrin, bifenazate, and etoxazole (17-20). Historically these mutations have been detected

84 through DNA sequencing that can be analyzed for the presence of the polymorphism. However, multiple

85 investigations have recently suggested using these polymorphisms as a diagnostic tool to inform

86 agricultural growers of the acaricide resistance status of individual field populations of T. urticae. This

87 would allow the grower to make an informed decision into which acaricide would adequately control

88 their local T. urticae populations and may prevent further resistance development (21,22). Three of the

89 most common polymorphisms studied are a mutation that results in the amino acid change. The first

90 results in a change of a glycine to a serine in the cytochrome b gene (G126S) in the mitochondrial

91 respiratory chain. This mutation has been shown to confer resistance to bifenazate, a carboxylic ester, in

92 T. urticae (23). The effects of the G126S mutation have recently been called into question and new

93 research suggests that this mutation cannot be used to predict acaricide resistance (24). Another

mutation at amino acid 1538 in the voltage gate channel gene results in a 94 phenylalanine changing to an

95 isoleucine. This mutation confers resistance to bifenthrin, a pyrethroid (16) and has also been shown to

96 confer resistance to the growth inhibitors, hexythiazox and clofentezine (25). Finally, a mutation in the

97 chitin synthase 1 gene that changes an isoleucine to a phenylamine has been shown to result in

98 resistance to etoxazole, a narrow spectrum systemic acaricide (20). These mutations have been

99 characterized by sequencing both susceptible and resistant populations of T. urticae and have provided

100 in-depth information on the development of resistance to acaricides in T. urticae (16-20, 23). A

101 restriction digest has also been used to screen mite populations for the I1017F mutation(26).”

In line 93, authors provide information that G126S substitution is not predictive of mite resistance to Bifenazate. But they do not provide such disclaimer for F1538I (Bifenthrin)!!

I understand that this research has been done while some of the relationships between SNAs and resistance status were not available. However, the paper is written NOW and has to position facts correctly.

7. PLOS authors have the option to publish the peer review history of their article (what does this mean?). If published, this will include your full peer review and any attached files.

Reviewer #2: **Yes: **Recep AY

Reviewer #3: No

Reviewer #4: **Yes: **Mohammad Shaef Ullah

Reviewer #5: No

Reviewer #6: **Yes: **Vojislava Grbic

---

## [Author Response · Author response to Decision Letter 1]

17 Nov 2022

Reviewer #4: 

The authors examined the use of a TaqMan qPCR-based approach to determine acaricide resistance genotypes in field-collected populations of T. urticae from peppermint fields and hop yards in the Pacific Northwest of the United States and confirmed the results with a multiplex targeted sequencing. Although the MS is improved much, the following point need to be addressed.

The authors changed the title as ‘Using targeted sequencing and TaqMan approaches to detect acaricide resistance associated SNPs in Tetranychus urticae collected from peppermint fields and hop yards’

I appreciate their decision and would suggest specifying the acaricide name(s) in the title rather acaricide.

We have changed the name of the manuscript to include the acaricides that we investigated.

Still, I am not convinced that without examining the median lethal dose assay of bifenthrin and etoxazole to detect the acaricide resistance associated SNPs. If this is not possible to perform these studies in this stage, that should be addressed in the discussion.

We thank the reviewer for their comment. We have further addressed this in the discussion of the manuscript, “In the future, all SNPs being considered for the detection of acaricide resistance using a TaqMan approach should be validated through LD50 assessments for each specific acaricides. In this investigation we were only able to conduct an LD50 assessment with bifenazate and, as such, further investigations should be conducted to confirm that the genotyping data matches resistance phenotypes for bifenthrin and etoxazole.”

Reviewer #6: 

I find that revisions did not address the main problem of this manuscript. Yes, authors demonstrated that TaqMan protocol can identify SNPs in mite populations, BUT, authors do not state clearly that we do not have genetic markers for mite pesticide resistance (except etoxazole based on the prior research). Instead, the revised manuscript states in the abstract, line 33: “Within this investigation we examined the use of a TaqMan qPCR-based approach to determine acaricide resistance genotypes in field-collected populations of T. urticae from peppermint fields and hop yards in the Pacific Northwest of the United States and confirmed the results with a multiplex targeted sequencing.” This is MISLEADING! Authors detected SNPs (genotypes), but not acaricide resistance.

It is not our intention to be misleading in any form. In the manuscript we address that SNPs have been classified through peer reviewed literature which have implicated them with resistance phenotypes. We have changed this section to read as, “Within this investigation we examined the use of a TaqMan qPCR-based approach to determine genotypes which have been previously associated with acaricide resistance in field-collected populations of T. urticae from peppermint fields and hop yards in the Pacific Northwest of the United States and confirmed the results with a multiplex targeted sequencing.”

Furthermore, in Introduction authors state, 

Line 81: SNPs in multiple, different genes have become associated with the development of resistance to different acaricide chemistries including bifenthrin, bifenazate, and etoxazole (17-20). Historically these mutations have been detected through DNA sequencing that can be analyzed for the presence of the polymorphism. However, multiple investigations have recently suggested using these polymorphisms as a diagnostic tool to inform agricultural growers of the acaricide resistance status of individual field populations of T. urticae. This would allow the grower to make an informed decision into which acaricide would adequately control their local T. urticae populations and may prevent further resistance development (21,22). Three of the most common polymorphisms studied are a mutation that results in the amino acid change. The first results in a change of a glycine to a serine in the cytochrome b gene (G126S) in the mitochondrial respiratory chain. This mutation has been shown to confer resistance to bifenazate, a carboxylic ester, in T. urticae (23). The effects of the G126S mutation have recently been called into question and new research suggests that this mutation cannot be used to predict acaricide resistance (24). Another mutation at amino acid 1538 in the voltage gate channel gene results in a phenylalanine changing to an isoleucine. This mutation confers resistance to bifenthrin, a pyrethroid (16) and has also been shown to confer resistance to the growth inhibitors, hexythiazox and clofentezine (25). Finally, a mutation in the chitin synthase 1 gene that changes an isoleucine to a phenylamine has been shown to result in resistance to etoxazole, a narrow spectrum systemic acaricide (20). These mutations have been characterized by sequencing both susceptible and resistant populations of T. urticae and have provided in-depth information on the development of resistance to acaricides in T. urticae (16-20, 23). A restriction digest has also been used to screen mite populations for the I1017F mutation(26).”

In line 93, authors provide information that G126S substitution is not predictive of mite resistance to Bifenazate. But they do not provide such disclaimer for F1538I (Bifenthrin)!!

We thank the reviewer for the comment and we want to make sure that the manuscript is accurately written and appreciate the feedback.

We have included the citation, “Riga M, Bajda S, Themistokleous C, Papadaki S, Palzewicz M, Dermauw W, Vontas J, Leeuwen TV. The relative contribution of target-site mutations in complex acaricide resistant phenotypes as assessed by marker assisted backcrossing in Tetranychus urticae. Scientific reports. 2017 Aug 23;7(1):1-2.” within the introduction and discussion. The discussion now reads: “The mutation F1538I, which has been previously associated with bifenthrin resistance, has also been suggested to be a poor predictor of resistance phenotypes (26). In the future, all SNPs being considered for the detection of acaricide resistance using a TaqMan approach should be validated through LD50 assessments for each specific acaricides. In this investigation we were only able to conduct an LD50 assessment with bifenazate and, as such, further investigations should be conducted to confirm that the genotyping data matches resistance phenotypes for bifenthrin and etoxazole.”

The reason why we choose to examine this mutation (F1538I) was the depth of information that implicate this mutation in resistance phenotypes in the Pacific Northwest. Including recent investigation:

Piraneo TG, Bull J, Morales MA, Lavine LC, Walsh DB, Zhu F. Molecular mechanisms of Tetranychus urticae chemical adaptation in hop fields. Scientific reports. 2015 Dec 1;5(1):1-2.

Wu M, Adesanya AW, Morales MA, Walsh DB, Lavine LC, Lavine MD, Zhu F. Multiple acaricide resistance and underlying mechanisms in Tetranychus urticae on hops. Journal of Pest Science. 2019 Mar;92(2):543-55.

Adesanya AW, Lavine MD, Moural TW, Lavine LC, Zhu F, Walsh DB. Mechanisms and management of acaricide resistance for Tetranychus urticae in agroecosystems. Journal of Pest Science. 2021 Jun;94(3):639-63.

I understand that this research has been done while some of the relationships between SNAs and resistance status were not available. However, the paper is written NOW and has to position facts correctly.

We appreciate the reviewers comment as we want to make sure that the manuscript is appropriate and accurately written, throughout the manuscript (abstract, introduction and discussion) we address new findings regarding G126S, including the findings of this research which further confirms Xue et al. 2021 research. We further have added a section regarding the ability to use F1538I as a predictor of resistance within the manuscript.

Line 41: While we were able to detect the SNPs associated with the examined acaricides, the mutation G126S was not an appropriate or accurate indicator for bifenazate resistance.

Line 93: The effects of the G126S mutation have recently been called into question and new research suggests that this mutation cannot be used to predict acaricide resistance (24). Another mutation at amino acid 1538 in the voltage gate channel gene results in a phenylalanine changing to an isoleucine. This mutation has been suggested to confer resistance to bifenthrin, a pyrethroid (16) and has also been shown to confer resistance to the growth inhibitors, hexythiazox and clofentezine (25). However, some studies bring into question the accuracy of F1538I to predict bifenthrin resistance (26).

Line 373: Recent studies agree with our findings and suggest that the G126S mutation may not be a good indicator that a population is resistant to bifenazate (24). It is likely that multiple markers will be required to accurately detect resistance in field populations and further that the G126S mutation is not an optimal indicator of resistance status. The mutation F1538I, which has been previously associated with bifenthrin resistance, has also been suggested to be a poor predictor of resistance phenotypes (26). In the future, all SNPs being considered for the detection of acaricide resistance using a TaqMan approach should be validated through LD50 assessments for each specific acaricides. In this investigation we were only able to conduct an LD50 assessment with bifenazate and, as such, further investigations should be conducted to confirm that the genotyping data matches resistance phenotypes for bifenthrin and etoxazole.

---

## [Decision Letter · Decision Letter 2]

25 Jan 2023

PONE-D-22-09155R2Using targeted sequencing and TaqMan approaches to detect acaricide (bifenthrin, bifenazate, and etoxazole) resistance associated SNPs in Tetranychus urticae collected from peppermint fields and hop yardsPLOS ONE

Dear Dr. Justin Clements,

Thank you for submitting your manuscript to PLOS ONE. After careful consideration, we feel that it has merit but does not fully meet PLOS ONE’s publication criteria as it currently stands. Therefore, we invite you to submit a revised version of the manuscript that addresses the points raised during the review process.

We look forward to receiving your revised manuscript.

Kind regards,

Islam Hamim, PhD

Academic Editor

PLOS ONE

Journal Requirements:

Reviewers' comments:

Reviewer's Responses to Questions

**Comments to the Author**

1. If the authors have adequately addressed your comments raised in a previous round of review and you feel that this manuscript is now acceptable for publication, you may indicate that here to bypass the “Comments to the Author” section, enter your conflict of interest statement in the “Confidential to Editor” section, and submit your "Accept" recommendation.

Reviewer #3: All comments have been addressed

Reviewer #5: All comments have been addressed

Reviewer #6: (No Response)

2. Is the manuscript technically sound, and do the data support the conclusions?

Reviewer #3: Yes

Reviewer #5: Yes

Reviewer #6: Yes

3. Has the statistical analysis been performed appropriately and rigorously? 

Reviewer #3: Yes

Reviewer #5: N/A

Reviewer #6: Yes

4. Have the authors made all data underlying the findings in their manuscript fully available?

Reviewer #3: Yes

Reviewer #5: Yes

Reviewer #6: (No Response)

5. Is the manuscript presented in an intelligible fashion and written in standard English?

Reviewer #3: Yes

Reviewer #5: Yes

Reviewer #6: Yes

6. Review Comments to the Author

Reviewer #3: This is the second revision of the original paper PONE-D-22-09155. Authors did a great job in the revision. They answered all questions raised by reviewers. It is ready to be published at this stage.

Reviewer #5: All my comments have been addressed. The manuscript can be considered for publication in PLOS One.

Reviewer #6: The effort to modify the manuscript is appreciated. However, authors have to go over manuscript and systematically change parts that refer to analysed SNPs as a tool for detection of pesticide resistance in mite populations. Authors state:” In the present study, we set out to investigate whether we could use a pooled TaqMan qPCR approach to monitor resistant populations of T. urticae. We chose to examine a limited set of three well defined markers G126S (Cytochrome b), F1538I (Voltage gated sodium channel), and I1017F (Chitin synthase 1) using a pooled TaqMan approach.” This is a good rationale for the study and is independent of utility of these SNPs for detection of mite pesticide resistance. Reading the manuscript, authors are continuously trying to justify the choice of SNPs and to link it to the resistance. Sorry for the insistence, but one should clearly state the purpose of the work and should critically state the utility of analysed SNPs as being diagnostic for mite pesticide resistance. Already in the field, there is confusion of whether they are usable or not. Authors themselves have chosen these SNPs based on initial publications that suggested that they can be diagnostic. This manuscript should not continue to propagate this misconception. For example running title: Detection of acaricide resistance in Tetranychus urticae using molecular based approaches” is not appropriate and should be changed. Also, authors introduced modifications as:” This mutation has been suggested to confer resistance to bifenthrin (please add REF), a pyrethroid (16) and has also been shown to confer resistance to the growth inhibitors, hexythiazox and clofentezine (25). However, some studies bring into question the accuracy of F1538I to predict bifenthrin resistance (26).” Can authors make a single statement in the introduction that will critically establish the utility of reported SNPs as diagnostic? The fact that most of them are not, is not diminishing their work.

Please have a careful reading of the manuscript and eliminate some grammar mistakes.

7. PLOS authors have the option to publish the peer review history of their article (what does this mean?). If published, this will include your full peer review and any attached files.

Reviewer #3: No

Reviewer #5: No

Reviewer #6: **Yes: **Vojislava Grbic

---

## [Author Response · Author response to Decision Letter 2]

8 Feb 2023

Reviewer 6:

1) The effort to modify the manuscript is appreciated. However, authors have to go over manuscript and systematically change parts that refer to analysed SNPs as a tool for detection of pesticide resistance in mite populations. Authors state:” In the present study, we set out to investigate whether we could use a pooled TaqMan qPCR approach to monitor resistant populations of T. urticae. We chose to examine a limited set of three well defined markers G126S (Cytochrome b), F1538I (Voltage gated sodium channel), and I1017F (Chitin synthase 1) using a pooled TaqMan approach.” This is a good rationale for the study and is independent of utility of these SNPs for detection of mite pesticide resistance. Reading the manuscript, authors are continuously trying to justify the choice of SNPs and to link it to the resistance. 

Sorry for the insistence, but one should clearly state the purpose of the work and should critically state the utility of analysed SNPs as being diagnostic for mite pesticide resistance. Already in the field, there is confusion of whether they are usable or not. Authors themselves have chosen these SNPs based on initial publications that suggested that they can be diagnostic.

We agree with the reviewer that the focus of the manuscript should be clearly stated through out the manuscript. We have clarified throughout the manuscript that we are specifically detecting resistance associated SNPs and not detecting resistance.

2) This manuscript should not continue to propagate this misconception. For example running title: Detection of acaricide resistance in Tetranychus urticae using molecular based approaches” is not appropriate and should be changed.

We have changed the running title from, “Detection of acaricide resistance in Tetranychus urticae using molecular based approaches” to, “Detection of acaricide resistance associated SNPs in Tetranychus urticae”

3) Also, authors introduced modifications as:” This mutation has been suggested to confer resistance to bifenthrin (please add REF), a pyrethroid (16) and has also been shown to confer resistance to the growth inhibitors, hexythiazox and clofentezine (25). However, some studies bring into question the accuracy of F1538I to predict bifenthrin resistance (26).” Can authors make a single statement in the introduction that will critically establish the utility of reported SNPs as diagnostic? The fact that most of them are not, is not diminishing their work.

We have added the following section to the introduction, “Additionally, recent findings indicate that some of the polymorphisms associated with resistance may not be useful as a diagnostic tool, including G126S in the cytochrome b gene and F1538I in the voltage gate channel (24,26).” We also hope to convey this point with the following statements in the discussion “In the future, all SNPs being considered for the detection of acaricide resistance using a TaqMan approach should be validated through LD50 assessments for each specific acaricides.” and “While these genetic approaches are constantly evolving based on our knowledge of putative and nearby polymorphisms (crucial for a stable assay), further work is needed to validate the importance of specific mutations and the metrics used for determining overall resistance profiles within a population before they can be used to inform pest management decisions.”

4) Please have a careful reading of the manuscript and eliminate some grammar mistakes.

The authors have gone through the manuscript for grammar and edited manuscript to the best of our ability.

---

## [Decision Letter · Decision Letter 3]

6 Mar 2023

Using targeted sequencing and TaqMan approaches to detect acaricide (bifenthrin, bifenazate, and etoxazole) resistance associated SNPs in Tetranychus urticae collected from peppermint fields and hop yards

PONE-D-22-09155R3

Dear Dr. Justin Clements

We’re pleased to inform you that your manuscript has been judged scientifically suitable for publication and will be formally accepted for publication once it meets all outstanding technical requirements.

Kind regards,

Islam Hamim, PhD

Academic Editor

PLOS ONE

Additional Editor Comments (optional):

Accepted.

Reviewers' comments:

Reviewer's Responses to Questions

**Comments to the Author**

1. If the authors have adequately addressed your comments raised in a previous round of review and you feel that this manuscript is now acceptable for publication, you may indicate that here to bypass the “Comments to the Author” section, enter your conflict of interest statement in the “Confidential to Editor” section, and submit your "Accept" recommendation.

Reviewer #3: (No Response)

Reviewer #6: All comments have been addressed

2. Is the manuscript technically sound, and do the data support the conclusions?

Reviewer #3: (No Response)

Reviewer #6: Yes

3. Has the statistical analysis been performed appropriately and rigorously? 

Reviewer #3: (No Response)

Reviewer #6: Yes

4. Have the authors made all data underlying the findings in their manuscript fully available?

Reviewer #3: (No Response)

Reviewer #6: Yes

5. Is the manuscript presented in an intelligible fashion and written in standard English?

Reviewer #3: (No Response)

Reviewer #6: Yes

6. Review Comments to the Author

Reviewer #3: Authors have adequately addressed my comments raised in a previous round of review. This manuscript is now acceptable for publication.

Reviewer #6: Thank you for making changes in the manuscript. Can you please make a change in the title in line 256: Targeted-sequencing and detection of bifenthrin (F1538I,) and etoxazole (I1017F) resistance. Please change resistance to polymorphisms.

7. PLOS authors have the option to publish the peer review history of their article (what does this mean?). If published, this will include your full peer review and any attached files.

Reviewer #3: No

Reviewer #6: **Yes: **Vojislava Grbic

---

## [Editor Report · Acceptance letter]

15 Mar 2023

PONE-D-22-09155R3 

Using targeted sequencing and TaqMan approaches to detect acaricide (bifenthrin, bifenazate, and etoxazole) resistance associated SNPs in Tetranychus urticae collected from peppermint fields and hop yards 

Dear Dr. Clements:

I'm pleased to inform you that your manuscript has been deemed suitable for publication in PLOS ONE. Congratulations! Your manuscript is now with our production department. 

Kind regards, 

on behalf of

Dr. Islam Hamim 

Academic Editor

PLOS ONE